# The French Early Breast Cancer Cohort (FRESH): A Resource for Breast Cancer Research and Evaluations of Oncology Practices Based on the French National Healthcare System Database (SNDS)

**DOI:** 10.3390/cancers14112671

**Published:** 2022-05-27

**Authors:** Elise Dumas, Lucie Laot, Florence Coussy, Beatriz Grandal Rejo, Eric Daoud, Enora Laas, Amyn Kassara, Alena Majdling, Rayan Kabirian, Floriane Jochum, Paul Gougis, Sophie Michel, Sophie Houzard, Christine Le Bihan-Benjamin, Philippe-Jean Bousquet, Judicaël Hotton, Chloé-Agathe Azencott, Fabien Reyal, Anne-Sophie Hamy

**Affiliations:** 1Residual Tumor & Response to Treatment Laboratory, RT2Lab, Translational Research Department, INSERM, U932 Immunity and Cancer, 75005 Paris, France; elise.dumas28@gmail.com (E.D.); beatriz.grandalrejo@curie.fr (B.G.R.); eric.daoud@curie.fr (E.D.); amyn.kassara@curie.fr (A.K.); floriane.jochum@curie.fr (F.J.); paul.gougis@curie.fr (P.G.); fabien.reyal@gmail.com (F.R.); 2INSERM, U900, 75005 Paris, France; chloe-agathe.azencott@mines-paristech.fr; 3MINES ParisTech, CBIO-Centre for Computational Biology, PSL Research University, 75006 Paris, France; 4Department of Surgical Oncology, Institut Curie, University of Paris, 75005 Paris, France; lucie.laot@gmail.com (L.L.); enora.laas@curie.fr (E.L.); sph.michel@gmail.com (S.M.); 5Department of Medical Oncology, Institut Curie, University of Paris, 75005 Paris, France; florencecoussy@hotmail.com; 6INRIA, DI/ENS, PSL Research University, 75006 Paris, France; 7Centre René Hughenin, Medical Oncology Department, 92210 Saint Cloud, France; alena.majdling@curie.fr (A.M.); rayan.kabirian@curie.fr (R.K.); 8Department of Gynecology, Strasbourg University Hospital, 67091 Strasbourg, France; 9Survey Data Science and Assessment Division, French National Cancer Institute (Institut National du Cancer INCa), 92100 Boulogne-Billancourt, France; shouzard@institutcancer.fr (S.H.); clebihan@institutcancer.fr (C.L.B.-B.); pjbousquet@institutcancer.fr (P.-J.B.); 10Inserm, IRD, SESSTIM, Equipe Labellisée Ligue Contre le Cancer, Aix-Marseille Université, 13005 Marseille, France; 11Department of Surgery, Institut Jean Godinot, 51100 Reims, France; judicael.hotton@reims.unicancer.fr; 12Institut Curie, PSL Research University, 75005 Paris, France

**Keywords:** breast cancer, National Health Data System, nationwide population, French database introduction

## Abstract

**Simple Summary:**

Because of an important disparity of care pathways and quality of care among women diagnosed with an early-stage breast cancer, we aimed to create a unique cohort of patients including all French women aged 18 years or over, treated by surgery and registered in the general health insurance coverage plan. After aggregating and annotating medico-administrative data on 235,368 early breast cancer patients, we open up perspectives for research on adverse effects, morbidity, mortality, the monitoring of care consumption, or medical-economic studies. We describe data sources, inclusion, and exclusion criteria, basic descriptive analyses, and longitudinal trends over time.

**Abstract:**

Background: Breast cancer (BC) is the most frequent cancer and the leading cause of cancer-related death in women. The French National Cancer Institute has created a national cancer cohort to promote cancer research and improve our understanding of cancer using the National Health Data System (SNDS) and amalgamating all cancer sites. So far, no detailed separate data are available for early BC. Objectives: To describe the creation of the French Early Breast Cancer Cohort (FRESH). Methods: All French women aged 18 years or over, with early-stage BC newly diagnosed between 1 January 2011 and 31 December 2017, treated by surgery, and registered in the general health insurance coverage plan were included in the cohort. Patients with suspected locoregional or distant metastases at diagnosis were excluded. BC treatments (surgery, chemotherapy, targeted therapy, radiotherapy, and endocrine therapy), and diagnostic procedures (biopsy, cytology, and imaging) were extracted from hospital discharge reports, outpatient care notes, or pharmacy drug delivery data. The BC subtype was inferred from the treatments received. Results: We included 235,368 patients with early BC in the cohort (median age: 60 years). The BC subtype distribution was as follows: luminal (80.2%), triple-negative (TNBC, 9.5%); *HER2*^+^ (10.3%), or unidentifiable (*n* = 44,388, 18.9% of the cohort). Most patients underwent radiotherapy (*n* = 200,685, 85.3%) and endocrine therapy (*n* = 165,655, 70.4%), and 38.3% (*n* = 90,252) received chemotherapy. Treatments and care pathways are described. Conclusions: The FRESH Cohort is an unprecedented population-based resource facilitating future large-scale real-life studies aiming to improve care pathways and quality of care for BC patients.

## 1. Introduction

Breast cancer (BC) is the most frequent cancer in French women, and the leading cause of cancer-related death in women. BC is treated by surgery, radiotherapy, chemotherapy, and endocrine therapy, which, together with the targeted therapies developed in recent decades, have greatly improved overall survival. Following the recent FDA approval of pembrolizumab, in combination with chemotherapy in the neoadjuvant setting [1], immunotherapies may be added to the therapeutic arsenal against BC in the near future.

The French national health insurance system covered 98.8% of the 67 million inhabitants of France in 2020 [2]. All the medical and administrative information relating to the reimbursement of French citizens for healthcare expenses are collected and aggregated in the National Health Data System (Système National des Données de Santé, SNDS) database [3]. In 2014, the French National Cancer Institute (Institut National du Cancer, INCa) set up the French cancer cohort, an exhaustive population-based cohort, based on SNDS data [4]. This resource aims to provide a robust and validated database for the analysis of cancer complications, adverse effects, morbidity, and mortality, and it provides opportunities for studying expenditure indicators and the monitoring of care consumption, quality and safety indicators, oncologic outcomes, geographic distributions, and care pathways. This resource currently contains amalgamated data for all sites of cancer, with no detailed data available separately for early BC.

We describe here a specific subset of the French cancer cohort comprising all women diagnosed with early breast cancer, in the context of the French Early Breast Cancer Cohort (FRESH). We describe data sources, inclusion and exclusion criteria, basic descriptive analyses, and longitudinal trends over time. We performed quality control and benchmarking against published data, and we discuss perspectives for improving oncology practices, generating research hypotheses, and overcoming existing challenges.

## 2. Materials and Methods

### 2.1. Data Source and Available Variables

Relevant data were identified with the Oncology Data Platform (ODP) available at the French National Cancer Institute (INCa). The ODP gathers together SNDS data for all individuals living in France with universal health insurance cover (98.8% of the population) [5] who were diagnosed with or treated for cancer between 2010 and 2018. The ODP has been described in detail elsewhere [4]. It includes (i) demographic data (sex, date of birth, zip code of the town of residence, vital status, date of death if appropriate, and health insurance regimen), (ii) hospital discharge reports (diagnoses, medical procedures, and expensive treatments), (iii) outpatient care (drugs dispensed, with the date of delivery, laboratory tests, and outpatient medical procedures) from the year preceding the date of the inclusion of the patient in the ODP up to 31 December 2018, and (iv) all long-term illness (LTI) records (diagnosis codes and date of disease onset) until 31 December 2018. The Diagnosis codes were recorded with the International Classification of Diseases—10th revision, ICD-10 [6]. Procedures were recorded with the CCAM classification (Classification Communes des Actes Médicaux). Molecules in outpatient care were fully identifiable and were recorded with CIP (Code Identifiant de Présentation) codes. In hospital, only costly innovative drugs part of a special reimbursement process called “list en sus” were recorded, under the form of UCD (Unités Communes de Dispensation) codes. Both the UCD and CIP codes were linked to the ATC classification (Anatomical Therapeutic and Chemical classification) of the World Health Organization. In France, several health insurance coverage plans exist, depending on occupational status. The general health insurance plan (“Régime Général”) gathers approximately 88% of the French population: employees in the industry, business, and service sectors; public service employees; and students.

### 2.2. Ethics and Data Protection

This study was conducted in the framework of a partnership between Institut Curie and the French INCa. It was performed in accordance with institutional and ethical rules concerning research using data from patients. The study was authorized by the French data protection agency (Commission nationale de l’informatique et des libertés—CNIL, under registration number 920017). No informed consent was required because the data used in the study was de-identified and re-used for research purposes, in accordance with French regulations applicable to the SNDS data.

### 2.3. Selection of the Patients

We applied 10 filters for patient inclusion in the cohort (Appendix A). We identified patients with (1) BC and (2) newly diagnosed between 1 January 2011 and 31 December 2017. We also applied sociodemographic filters to exclude: (3) male patients, as they represent a very specific population with different therapeutic approaches [7]), (4) patients under the age of 18 years at inclusion, and (5) patients who were not registered in the general health insurance coverage plan (ensuring exhaustivity for vital status outcomes). Patients who did not undergo breast surgery in the year following inclusion (6) were excluded from the cohort to ensure the exclusion of relapses of a previous BC diagnosis. Such exclusion criteria was unlikely to bias significantly the population cohort, as more than 95% of patients treated for an incident non-metastatic BC in France received surgery [8]. It also enabled fixing an index surgery date for use as a reference for the definition and the settings (neoadjuvant vs. adjuvant) of the other BC treatments. We excluded patients with other concomitant cancers (7) to ensure that the retrieved cancer treatments are BC treatments. Patients with evidence of a previous BC, (8) or evidence of stage IV metastatic BC at diagnosis, (9) were also excluded from the cohort (metastatic BC is an incurable disease with specific chemotherapy and targeted therapy molecules, divergences in medical opinions regarding need for surgery [9] and a low median survival of about 3 years [10]). Finally, we excluded patients with poor-quality or inconsistent data (10). The full process of patient selection is detailed in the Appendix A.

### 2.4. BC Treatments

#### 2.4.1. Surgery

Surgery for BC was tagged with CCAM procedure codes (Appendix A) at the hospital, with classification into five categories: (1) mastectomy with axillary surgery, (2) mastectomy without axillary surgery, (3) partial mastectomy with axillary surgery, (4) partial mastectomy without axillary surgery, and (5) axillary surgery without breast surgery. The index surgery for BC was defined as the date on which the first breast surgical operation for BC (categories 1 to 4 above) took place, in the year of inclusion or the following year. This date was used as a reference for the definition of the other treatments. The decision rules used to bin breast surgery types (partial mastectomy vs. mastectomy) and axillary surgery (yes vs. no) are detailed in the Appendix A, together with the rules for data handling for patients undergoing several surgical procedures.

#### 2.4.2. Radiotherapy

Radiotherapy (RT) sessions were identified by ICD-10 diagnosis codes or CCAM procedure codes (Appendix A). We used temporal restrictions to exclude RT sessions for another cancer or a BC relapse: a patient was considered to have been treated with RT if she had at least one RT session between 150 days before and up to 365 days after BC index surgery. Thresholds have been set in accordance with clinical practices. Decision rules concerning the radiotherapy setting (neoadjuvant/adjuvant/both) and the identification of start and end dates are detailed in the Appendix A.

#### 2.4.3. Chemotherapy

Chemotherapy (CT) sessions were identified by ICD-10 diagnosis codes, CCAM procedure codes and ATC molecule codes for hospital and outpatient care (Appendix A). A patient was considered to have been treated with CT if she had at least one CT session between 250 days before and up to 180 days after BC index surgery, in accordance with clinical practices. Decision rules for chemotherapy setting (neoadjuvant/adjuvant/both) and the identification of start and end dates, and intervals between CT sessions are detailed in the Appendix A.

The following seven CT regimens were identified: (1) anthracyclines, (2) anthracyclines/docetaxel, (3) anthracyclines/paclitaxel, (4) docetaxel, (5) paclitaxel, (6) unknown or (7) other; as detailed in the Appendix A, together with the number of cycles, and displayed in Appendix A. Regimens containing paclitaxel were fully identifiable, whereas regimens containing only anthracyclines and/or docetaxel were not and were mostly tagged as “Unknown”.

#### 2.4.4. Endocrine Therapy

Endocrine therapy (ET) intake was tagged on the basis of the outpatient delivery of (1) tamoxifen, (2) aromatase inhibitors (AI), or (3) gonadotropin-releasing hormone agonists (GnRH agonists), identified with ATC codes (Appendix A). Decision rules for endocrine therapy setting (neoadjuvant followed by adjuvant/adjuvant), for the exclusion of treatments linked to fertility preservation procedures, and for the identification of start and end dates are detailed in the Appendix A.

ET regimens were classified into seven categories according to the ET molecules delivered over the whole study period: (1) tamoxifen only (tamoxifen); (2) at least one delivery of tamoxifen plus at least one delivery of GnRH agonists (tamoxifen with GnRH agonists); (3) at least one delivery of tamoxifen followed by at least one delivery of AI (tamoxifen followed by AI); (4) AI only (AI); (5) at least one delivery of AI and at least one delivery of GnRH agonists (AI with GnRH agonists); (6) at least one delivery of AI followed by at least one delivery of tamoxifen (AI followed by tamoxifen); (7) all other cases, including the delivery of GnRH agonists alone, the delivery of three types of ET, or multiple sequential combinations of AI and tamoxifen (others).

#### 2.4.5. Targeted Therapy (TT)

Only two targeted therapies (TT) (trastuzumab and pertuzumab) have been approved for the early BC setting. *Anti-HER2 (human epidermal growth factor receptor 2)* targeted therapy (TT) sessions were identified by ATC codes (Appendix A) for trastuzumab and/or pertuzumab. Patients were considered to have received targeted therapy if they had at least one anti-*HER2* targeted therapy session between 250 days before and up to 180 days after BC index surgery, in accordance with clinical practices. The decision rules for TT setting (neoadjuvant followed by adjuvant/adjuvant) and the identification of start and end dates are detailed in the Appendix A. TT regimens were classified as: (1) trastuzumab alone (trastuzumab) or (2) pertuzumab with or without trastuzumab (pertuzumab +/− trastuzumab). The decision rules for combinations of TT with other systemic treatments and the number of cycles are detailed in the Appendix A and displayed in Appendix A. Of note, the diagnostic code of chemotherapy sessions does not enable distinguishing sessions of TT combined with chemotherapy and sessions of TT alone.

Combinations of TT and systemic treatments were classified as follows: (1) Anthracycline-based regimen alone and then a combination of docetaxel and TT (anthracyclines/docetaxel-TT); (2) anthracycline-based regimen alone and then a combination of paclitaxel and TT (anthracyclines/paclitaxel-TT); (3) combination of a docetaxel-based regimen and TT (docetaxel-TT); (4) combination of paclitaxel and TT, as described by Tolaney et al. [11], (paclitaxel-TT (Tolaney)); (5) combination of endocrine therapy and TT, without chemotherapy (TT-ET); and (6) any other combinations including TT (other).

### 2.5. Date of First BC Treatment

We defined the date of first BC treatment as the date of BC index surgery or the start date of neoadjuvant chemotherapy (NAC), neoadjuvant endocrine therapy (NET), neoadjuvant radiotherapy (NRT) or neoadjuvant anti-*HER2* targeted therapy (NTT), whichever occurred first.

### 2.6. BC Diagnostic Procedures and Date of BC Diagnosis

Procedures for BC diagnosis were identified by CCAM procedure codes in the hospital and outpatient settings (Appendix A) and were classified into three categories: breast core biopsy, fine-needle aspiration cytology, and breast imaging procedures, including mammography, mammary ultrasound, mammary MRI, CT scan, and galactography. If at least one breast core biopsy had been performed in the 12 months preceding the date of first BC treatment, we considered the diagnosis of BC to be set by the earliest breast core biopsy within this time range. Otherwise, we considered it to be set by the earliest breast fine-needle cytology aspiration. If neither breast core biopsy nor fine-needle aspiration cytology had been performed, the diagnosis was assumed to have been based on a breast imaging procedure. In such cases, the BC diagnosis date was set as the date of the earliest of such procedures, starting from the breast imaging procedure closest to the date of first BC treatment and going back breast imaging procedures one by one, as long as they were separated by no more than a month interval. The date of first BC treatment was taken as the date of diagnosis if no diagnostic procedure was recorded.

### 2.7. Variables of Interest

#### 2.7.1. Sociodemographic Variables

Age at diagnosis was calculated by the rounded difference, in years, between the date of BC diagnosis and the date of birth.

#### 2.7.2. Inferred BC Subtype

No information about the histological characteristics of the tumor was available from the database. We therefore inferred BC subtype from the treatments received. The tumors of patients receiving anti-*HER2* TT were classified as *HER2*^+^. Within this category, the tumors of patients receiving ET were classified as *HER2*^+^/HR^+^, and those of patients not receiving ET were classified as *HER2*^+^/HR^−^. The tumors of patients who received ET without anti-*HER2* TT were considered to be luminal. The tumors of patients receiving chemotherapy with no ET or anti-*HER2* TT were classified as triple-negative breast cancers (TNBCs). Finally, the tumors of patients treated exclusively by surgery with or without radiotherapy were considered to have an undefined subtype.

#### 2.7.3. Nodal Status

Lymph node involvement was tagged by the presence of at least one ICD-10 diagnosis code for node disease (C773) between 250 days before and up to 180 days after BC index surgery.

### 2.8. Statistical Analysis

Analyses were performed with R software, version 3.6.3. The study population was described in terms of frequencies for qualitative variables, or medians and interquartile ranges (IQR) for quantitative variables. For variables suspected to have a multimodal distribution based on graphical assessment, we performed a statistical test based on the critical bandwidth statistic with the modetest function of the R package multimode [12]. If the null hypothesis of unimodality was rejected in statistical tests, the number of modes was assessed graphically, and the position of the modes was inferred by kernel density estimation with gaussian kernels (R multimode package; function locmodes). The threshold for statistical significance was set at *p* = 0.05. We prevented undesirable edge effects, by restricting figures with a continuous variable (age, date of diagnosis, chemotherapy start date, etc.) on the *x*-axis to strata of the continuous variable (month–years for dates, integer units for age) containing at least 50 patients.

Overall survival (OS) was defined as the time, in months, from BC index surgery to death or to 1 March 2019, whichever occurred first. Data for patients still alive on 1 March 2019 were censored at this date. Median follow-up and its interquartile range (IQR) were assessed by reverse Kaplan-Meier methods. Unadjusted survival probabilities were estimated by the Kaplan-Meier method, and survival curves were compared in log-rank tests.

## 3. Results

### 3.1. Patients and Tumor Characteristics

#### 3.1.1. Age at BC Diagnosis

We included 235,368 women from the 455,711 patients with a diagnosis code of BC identified in the final cohort (Appendix A). The characteristics of the patients are summarized in Table 1. Median age at diagnosis was 60 years (Figure 1A).

The age distribution of the patients was bimodal, with two incidence peaks, at 50.3 and 65.0 years, respectively. The value of the second mode of the distribution increased steadily over time, from 62.7 years in 2011 to 68.0 years in 2017 (Appendix A).

#### 3.1.2. Mode of Diagnosis

A procedure for pathology analysis was performed before treatment in 93.3% of patients (biopsy *n* = 214,874, 91.3%; cytology *n* = 4808, 2.0%) (Appendix A). Imaging was the sole diagnostic procedure before treatment in 5.6% of patients (*n* = 13,250), and 1.0% of the patients underwent no diagnostic procedures at all before treatment (*n* = 2436). This absence of diagnostic procedures was more marked at extreme ages, in the youngest and oldest women (Appendix AB,C), and in patients whose first treatment was surgery (Appendix AD–F) than in patients receiving neoadjuvant treatment (Appendix AG–I).

#### 3.1.3. Inferred BC Subtypes

It was possible to infer the BC subtype in 190,980 (81.1%) patients, whereas the BC subtype was undefined for 44,388 patients (18.9% of the total cohort) (Appendix A). The distribution of inferred BC subtypes was as follows: luminal (80.2%), TNBC (9.5%), and *HER2*^+^ (10.3%) (Figure 1B). The proportion of TNBC and *HER2*^+^ BC decreased with advancing age (Figure 1C,D). The bimodal distribution of BC cases by age was also observed for the luminal and undefined subtypes, but not for the TNBC, *HER2*^+^/HR^+^, and the *HER2*^+^/HR^−^ subsets (Figure 1E–H and Appendix A).

#### 3.1.4. Nodal Status

Lymph node involvement was present in 18.8% of the cases (*n* = 44,204) (Appendix AA) and varied with age (Appendix AB) and inferred BC subtype (Appendix AC). For the patients for whom a BC subtype was inferred, the proportion of node-positive tumors was lowest for luminal BCs (21.5%), and highest for *HER2*^+^/HR^−^ BCs (30.1%).

### 3.2. BC Treatments

#### 3.2.1. Sequence of Treatments and Care Pathways

In accordance with the inclusion criteria, 100% of the patients underwent surgery (*n* = 235,368) (Figure 2A).

The distribution of other BC treatments was as follows: radiotherapy (*n* = 200,685, 85.3%), endocrine therapy (*n* = 165,655, 70.4%), chemotherapy (*n* = 90,252, 38.3%), and targeted therapy (*n* = 19,722, 8.4%).

Four main care pathways, accounting for 99.9% of patients, were identified: (1) surgery without chemotherapy (*n* = 143,042, 60.8%); (2) surgery followed by chemotherapy (*n* = 72,659, 30.9%); (3) neoadjuvant chemotherapy (*n* = 17,313, 7.4%); (4) neoadjuvant endocrine therapy (*n* = 2188, 0.9%) (Figure 2B). The relative distribution of the four main care pathways is shown in Figure 2C.

The proportion of patients treated with either neoadjuvant chemotherapy or adjuvant chemotherapy decreased with advancing age, whereas the proportion of patients treated with neoadjuvant ET increased with advancing age (Figure 2D).

#### 3.2.2. Locoregional Treatments

Surgery:

The distribution of surgical procedures was as follows: 73.6% of patients underwent partial mastectomy and 26.4% underwent mastectomy (Appendix AA–D); 83.8% of patients underwent axillary surgery, and 16.2% did not (*n* = 38,231) (Appendix AE). The type of surgical procedure varied with age, BC subtype, and nodal status (Appendix AB,D,F–H).

Radiotherapy:

Most patients underwent radiotherapy (85.3%) (Appendix AA), and the rate of radiotherapy varied with age, BC subtype, and nodal status (Appendix AB–D). This rate was higher in patients treated by partial mastectomy than in patients undergoing full mastectomy (93.3% vs. 63.0%) (Appendix AE–H).

#### 3.2.3. Systemic Treatments

Chemotherapy:

About one third (38.3%) of the study population received chemotherapy (Appendix AA), and the rate of chemotherapy varied with age, BC subtype, and nodal status (Appendix AB–D). Chemotherapy was administered in the neoadjuvant (*n* = 15,627, 17.3%), adjuvant (*n* = 72,939, 80.8%), or both settings (*n* = 1686, 1.9%) (Appendix AE–H).

Chemotherapy regimens could be inferred in 55.0% of cases (*n* = 50,559) but were unknown for 45.0% of cases (*n* = 41,379) (Figure 3A). 

The use of paclitaxel-only regimens increased with advancing age (Figure 3B,C). Most patients received six cycles of chemotherapy (51,517, 60.1%), and the proportions of patients receiving four (9966, 11.6%) or eight cycles (9405, 11.0%) were similar (Figure 3D).

The number of cycles depended on age at BC diagnosis (Figure 3E,F) and chemotherapy setting (Appendix AA–F).

#### 3.2.4. Targeted Therapy

In total, 19,722 (8.4%) patients received targeted therapy, more frequently in the adjuvant setting than in the neoadjuvant, followed by the adjuvant setting (77.6% vs. 22.4%, respectively; Appendix AA,B) and more frequently with trastuzumab alone than with pertuzumab +/− trastuzumab (97.8% vs. 2.2%, respectively; Appendix AC,D).

TT was mostly combined with an anthracycline/docetaxel-based regimen (*n* = 8697, 44.1%) (Appendix AE). The use of this regimen decreased with advancing age, in parallel with an increase in paclitaxel-TT (Tolaney) and TT associated with endocrine therapy (no chemotherapy) (Appendix AF,G).

#### 3.2.5. Endocrine Therapy

In total, 165,655 patients (70.4%) received endocrine therapy, mostly in the adjuvant setting (Figure 4A–D).

The three principal ET regimens were (1) AI (*n* = 102,757, 62.0%), (2) tamoxifen (*n* = 35,187, 21.2%), and (3) tamoxifen followed by AI (*n* = 10,350, 6.2%), and the type of ET regimen depended strongly on age (Figure 4F,G).

### 3.3. Trends over Time

The trends in treatments or pathways over time are displayed in Appendix A. The proportion of patients undergoing a pathology procedure for diagnostic purposes before treatment increased over the study period (Appendix AA,B). The type of breast surgery did not vary much over time (Appendix AC,D). The proportion of patients treated with NAC increased slightly, the proportion of patients treated with adjuvant CT decreased, and the proportion of patients without CT increased (Appendix AE,F). ET, CT, and TT combination regimens changed significantly over time (Appendix AG–L).

### 3.4. Survival Outcomes

The median follow-up was 54.6 months (IQR: 33.9; 75.6), and 15,503 patients died (6.6%). Death was significantly associated with age at BC diagnosis (Figure 5A), inferred BC subtype (Figure 5B), and nodal status (Figure 5C).

## 4. Discussion

In this study of 235,368 French women newly diagnosed with early BC, we constitute one of the largest national cohorts of BC patients treated within a universal healthcare system described to date. This resource will be very useful for a number of reasons.

First, the SNDS is one of the largest exhaustive nationwide aggregated health data resources worldwide [4]. Several other large databases of BC patients exist worldwide. The Surveillance, Epidemiology, and End Results (SEER) program of the National Cancer Institute collects data on cancer diagnoses, treatment, and survival for approximately 30% of the United States (US) population (Duggan et al., 2016). The National Cancer Database (NCDB) has amassed more than 34 million hospital records for cancer patients [13] and contains data for patients from the United States who received any element of their cancer care as part of a cancer program accredited by the American College of Surgeons Commission of Cancer (CoC). The NCCN Breast Cancer Outcomes Database (DB) contains data for patients receiving all or some of their treatment at one of eight US reporting centers [14,15]. The limitations of these databases include a lack of exhaustivity. There may also be biases, as the NCDB and NCCN Breast Cancer Outcomes DB are not population-based databases and only consider the care of those who had access to and received treatment at major academic cancer centers. Such biases can be ruled out in the exhaustive population-based registries of northern European countries such as Sweden, Norway, Finland, and Denmark [16,17,18,19]. We report here a much larger cohort of BC patients, which could be used for studies with high statistical power.

One major challenge in the use of reimbursement databases is the accuracy with which the reimbursement indicators reflect the medical condition. Over the last decade, the accuracy and reliability of SNDS have been evaluated for a very large range of medical conditions, in patients with prosthetic heart valves [20], in patients with Crohn’s disease [21], in parturient women [22], or for risks associated with exposure to certain treatments, such as cyproterone acetate [23], statins [24], thiopurines, or TNF antagonists [25]. Several studies have been performed in the BC field. Algorithms for identifying incident cancer cases in French administrative health databases have been published [26,27]. Other studies have described the care pathways of BC patients [28,29], the compliance with endocrine therapy [30,31], or the risk of hematologic malignancies following the use of G-CSF [32]. There have also been studies focusing on particular conditions in specific subpopulations, such as BC in male patients [33].

In our cohort, the distributions of the proxies for BC we describe are consistent with those from previous studies. We excluded 4.5% of the population due to a suspicion of stage IV BC at diagnosis, consistent with published rates, which are usually estimated at about 5% [34,35]. The rate of nodal involvement was also similar to the value of about 25% for the SEER data [36], and the increase in risk from luminal to TNBC to *HER2*^+^ BCs was also similar [37]. We also report decreasing proportions of TNBC and *HER2*^+^ BCs with advancing age, consistent with epidemiological evidence [38,39]. Our data are also consistent with the trends in clinical practice over time. The proportion of paclitaxel–trastuzumab regimens increased significantly after 2015, following the publication of Tolaney’s work [11], which introduces the use of adjuvant paclitaxel plus trastuzumab in small node-negative *HER2*^+^ BCs. Similarly, seven fatal cases of toxic enterocolitis occurred in France in 2016 and were suspected to be linked to docetaxel. In response, docetaxel was temporarily banned following warnings issues by the French national agency for medicines and health product safety (ANSM), and our data show a dramatic increase in the use of paclitaxel-based CT regimens as an alternative to docetaxel since January 2017.

The strengths of our study include the collection of data for an unprecedented number of BC patients. The use of reimbursement data are particularly appropriate in this setting because inferences can be made about BC biology due to the specificity of treatments, such as endocrine therapy and anti-*HER2* therapies targeting particular molecular alterations.

Conversely, we cannot exclude the possibility of a selection bias because we applied stringent criteria for patient inclusion, to ensure that only patients with high-quality data were retained. The identification of patterns of treatment based on codes (ICD-10, CCAM etc.) may also be subject to coding errors or discrepancies in coding methods between centers. Finally, BC subtypes were defined on the basis of treatment, making it impossible to classify patients with HR^+^ cancers who refused ET or patients with *HER2*^+^ BCs that were not treated with targeted therapies.

The FRESH cohort opens up multiple perspectives. At a patient level, such a resource could be used to analyze outpatient care during and after treatment, to identify rare adverse events, to monitor early complications of treatment and patterns of late sequalae, or to assess the impact of breast cancer on psychosocial status and quality of life. At a cancer care center level, benchmarks could be established with performance or quality indicators, such as EUSOMA [40], and real-time monitoring of care could be implemented in the context of continually changing guidelines. At a national level and from an economic standpoint, this cohort could be used to perform medico-economic studies, and to rationalize healthcare expenditure. Finally, in terms of research and development, this cohort represents a strategic opportunity for generating hypotheses and optimizing the potential for innovation to improve cancer care.

## 5. Conclusions

The FRESH cohort gathers structured medical and administrative data of nearly all French women with early-stage BC newly diagnosed between 2011 and 2017 (*n* = 235,368). It is an unprecedented population-based resource facilitating future large-scale real-life studies and aiming to improve care pathways and quality of care for BC patients.

## Figures and Tables

**Figure 1 cancers-14-02671-f001:**
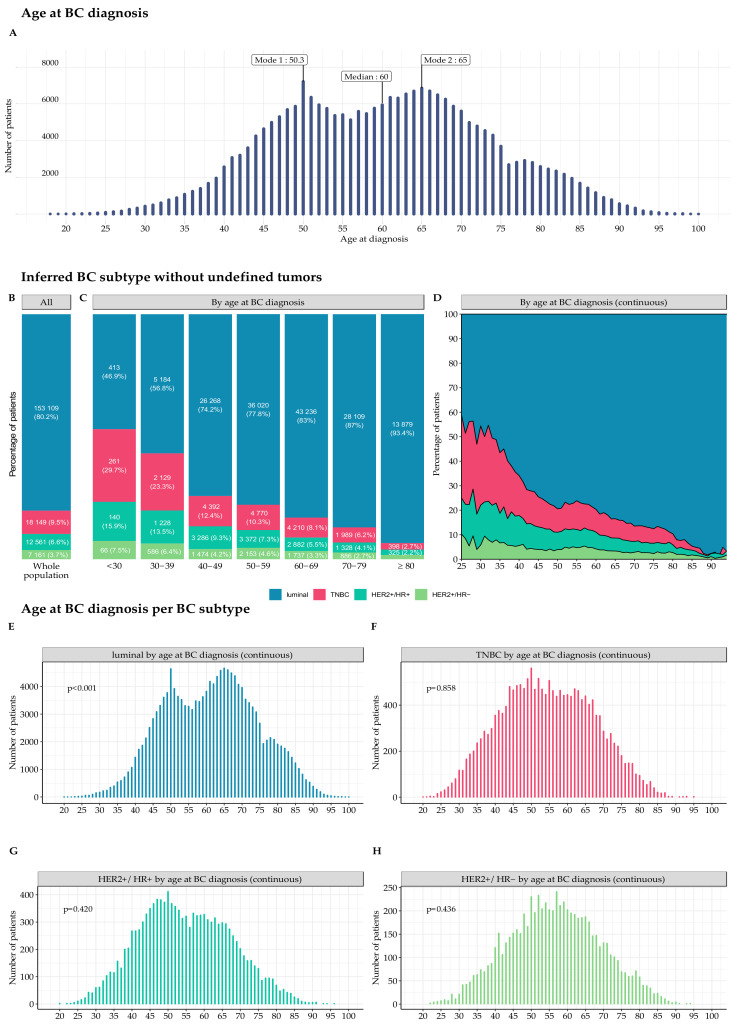
Age at BC diagnosis and inferred BC subtype, by age at BC diagnosis, excluding undefined tumors. (**A**) Number of patients included in the FRESH cohort, by age at BC diagnosis. The age distribution is bimodal, with two inferred incidence peaks at 50.3 and 65.0 years (*p*-value for non-unimodality < 0.001). Median age is 60 years; (**B**) Inferred BC subtype percentages for the whole population, excluding undefined tumors (*n* = 190,980); (**C**) Inferred BC subtype percentages per age class at BC diagnosis, excluding undefined tumors. Raw figures for subgroups representing less than 2% of the corresponding age class are not displayed on the graph, to ensure readability. For the group ≥80 years old: *n* = 259 (1.7%) for the *HER2*^+^/HR^−^ group; (**D**) Inferred BC subtype percentage by age at BC diagnosis, excluding undefined tumors. The cohort is restricted to patients aged from 25 to 94 years (*n* = 190,816); (**E**) Age distribution of patients with an inferred luminal subtype tumor (*n* = 153,109) at BC diagnosis (*p*-value for non-unimodality < 0.001); (**F**) Age distribution of patients with an inferred TNBC subtype tumor (*n* = 18,149) at BC diagnosis (*p*-value for non-unimodality = 0.858); (**G**) Age distribution of patients with an inferred *HER2*^+^/HR^+^ subtype tumor (*n* = 12,561) at BC diagnosis (*p*-value for non-unimodality = 0.420); and (**H**) Age distribution of patients with an inferred *HER2*^+^/HR^−^ subtype tumor (*n* = 7161) at BC diagnosis (*p*-value for non-unimodality = 0.436). Abbreviations: BC = breast cancer; HR^+^ = hormone receptor-positive; HR^−^ = hormone receptor-negative; TNBC = triple-negative breast cancer subtype; and *HER2*^+^ = human epidermal growth factor receptor 2-positive.

**Figure 2 cancers-14-02671-f002:**
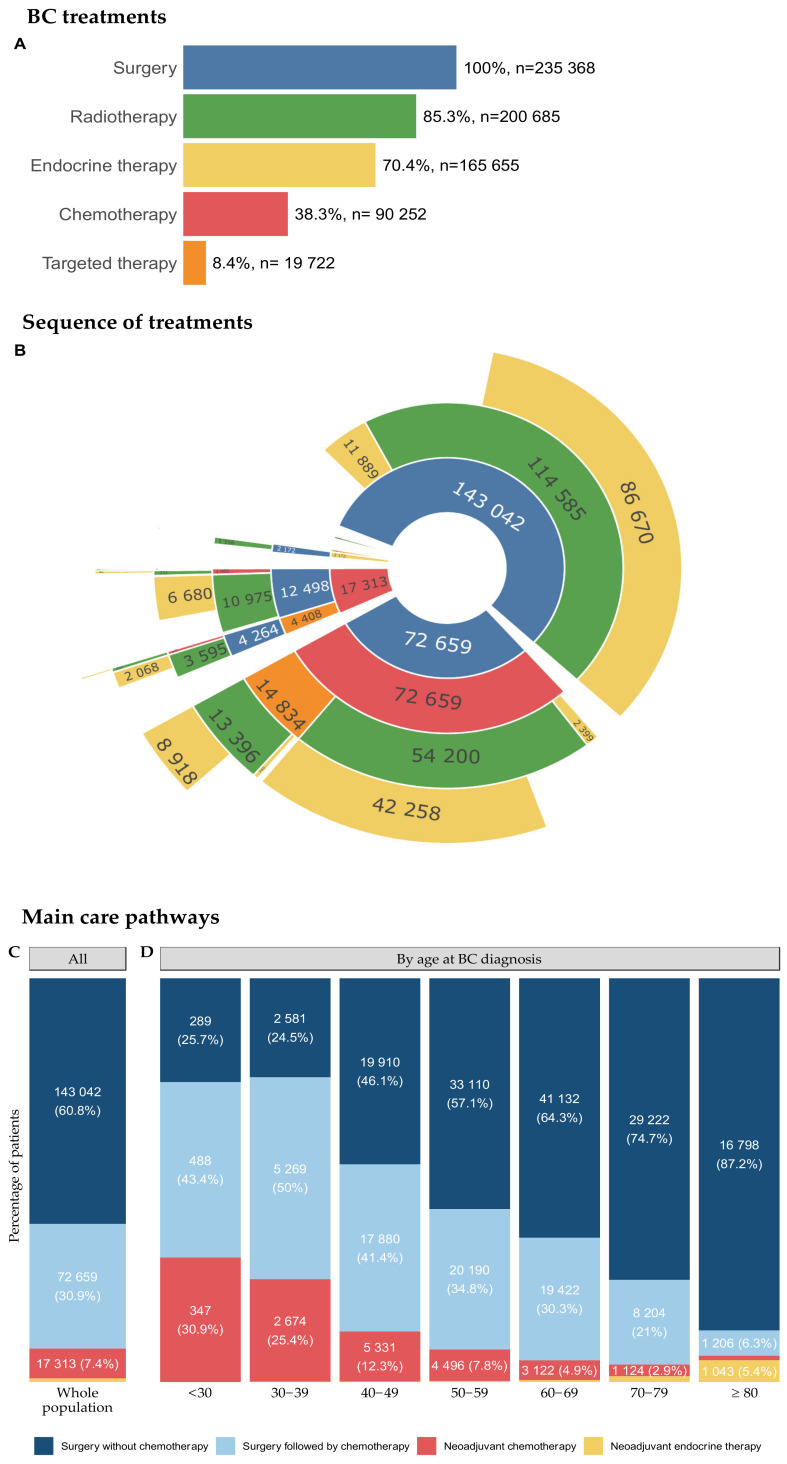
BC treatment, sequence of treatments and main care pathways, by age at BC diagnosis. (**A**) Distribution of BC treatment. (**B**) Sequential care pathways. Care pathways are displayed from inwards to outwards. For instance, *n* = 143,042 patients received surgery first without chemotherapy. Among those, surgery was followed by radiotherapy for *n* = 114,585 patients and by endocrine therapy without radiotherapy for *n* = 11,889. Treatment sequences are displayed in the following order: neoadjuvant chemotherapy (NAC)-neoadjuvant targeted therapy (NTT)-neoadjuvant radiotherapy (NRT)-neoadjuvant endocrine therapy (NET)-surgery-adjuvant chemotherapy–adjuvant targeted therapy–adjuvant radiotherapy–adjuvant endocrine therapy. The continuation of NET or NTT after surgery is not considered to constitute adjuvant endocrine therapy or adjuvant targeted therapy; (**C**) Distribution of the four main care pathways extracted from (**B**): (i) surgery without chemotherapy, (ii) surgery followed by chemotherapy, (iii) neoadjuvant chemotherapy (NAC), (iv) neoadjuvant endocrine therapy (NET). For the sake of clarity, radiotherapy is not spelled out in trajectories. The vast majority of the patients in the cohort (85.3%) underwent radiotherapy as part of their care pathway. Patients with neoadjuvant chemotherapy are classified as NAC irrespective of their NRT/NET status. Patients with NRT and NET are classified as NRT (*n* = 166). This category is so rare that it is not displayed on the plot. As a consequence, the patients in the NET group received neoadjuvant endocrine therapy and neither NRT nor NAC. Targeted therapy is always given in combination with either chemotherapy or endocrine therapy. This implies that NTT patients are in the NAC, NRT, or NET group. Raw data for subgroups representing less than 2% of the total population are not displayed on the graph, to ensure readability: 2169 (0.9%) patients are in the NET group; (**D**) Distribution of the four main treatment trajectories extracted from (**B**) per age class at BC diagnosis. Raw data for subgroups representing less than 2% of the corresponding age class are not displayed on the graph, to ensure readability. The values per age class are: for the <30 year-old group: *n* < 10; 30–39 years old: *n* = 25 (0.2%); 40–49 years old: *n* = 92 (0.2%); 50–59 years old: *n* = 207 (0.3%); 60–69 years old: *n* = 355 (0.6%); and 70–79 years old: *n* = 610 (1.6%) for the NET group. For the ≥80 years age class: *n* = 215 (1.1%) for the NAC group. Abbreviations: BC = breast cancer; CT = chemotherapy; NAC = neoadjuvant chemotherapy; NET = neoadjuvant endocrine therapy; NTT = neoadjuvant targeted therapy; and NRT = neoadjuvant radiotherapy.

**Figure 3 cancers-14-02671-f003:**
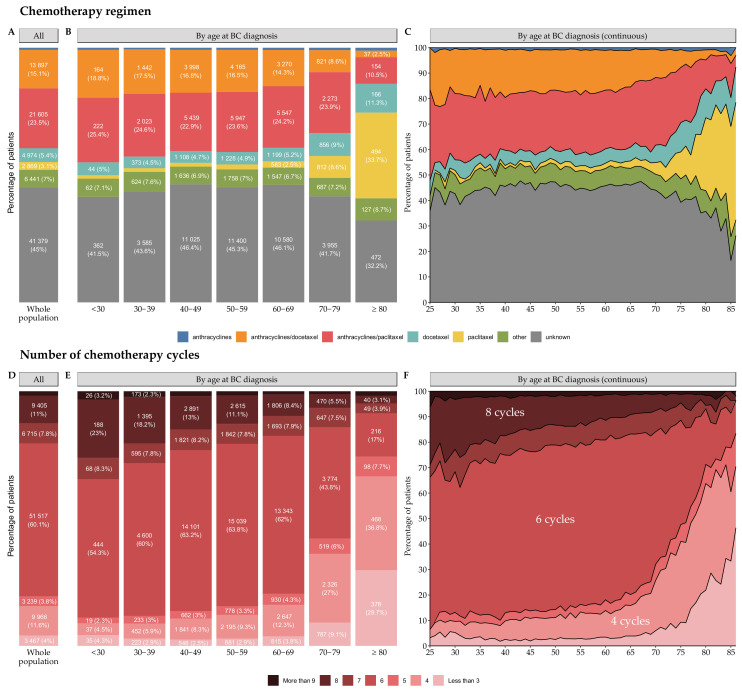
Chemotherapy regimen and number of cycles, by age at diagnosis. (**A**) Chemotherapy regimen for the total population (*n* = 91,938). Raw data for subgroups representing less than 2% of the total population are not displayed on the graph to ensure readability. In the anthracyclines group, *n* = 773 (0.8%); (**B**) Chemotherapy regimen, by age class at BC diagnosis. Raw data for subgroups representing less than 2% of the corresponding age class are not displayed on the graph, to ensure readability. The values are: for the <30 years old class: *n* = 12 (1.4%), 30–39 years old: *n* = 122 (1.5%), 40–49 years old: *n* = 361 (1.5%), 50–59 years old: *n* = 485 (1.9%) for the paclitaxel group. In the <30 years old class: *n* < 10, 30–39 years old: *n* = 49 (0.6%), 40–49 years old: *n* = 203 (0.9%), 50–59 years old: *n* = 205 (0.8%), 60–69 years old: *n* = 202 (0.9%), 70–79 years old: *n* = 90 (0.9%), and ≥80 years old: 17 (1.2%) for the anthracyclines group; (**C**) Chemotherapy regimen, by age at BC diagnosis. The cohort is restricted to patients aged from 25 to 86 years (*n* = 91,750); (**D**) Number of chemotherapy cycles for the total population. The number of chemotherapy cycles was calculated by setting (*n* = 85,752). A patient with six cycles of neoadjuvant chemotherapy followed by four cycles of adjuvant chemotherapy was counted as having both six cycles and four cycles of treatment. Settings with missing numbers of chemotherapy cycles are not displayed (*n* = 6186). Raw data for subgroups representing less than 2% of the total population are not displayed on the graph, to ensure readability: 1443 (1.7%) settings are in the more than 9 cycles group; (**E**) Number of chemotherapy cycles by age class (*n* = 85,752) at BC diagnosis. Raw data for subgroups representing less than 2% of the corresponding age class are not displayed on the graph, to ensure readability. The values are: for the 40–49 years old class: *n* = 441 (2.0%), 50–59 years old: *n* = 412 (1.7%), 60–69 years old: *n* = 271 (1.3%), 70–79 years old: *n* = 98 (1.1%), and ≥80 years old: *n* = 22 (1.7%) for the more than 9 cycles group; and (**F**) Chemotherapy regimens by age at BC diagnosis. The cohort is restricted to patients aged from 25 to 86 years (*n* = 85,589). Abbreviations: BC = breast cancer, More than 9 = more than 9 cycles, 8 = 8 cycles, 7 = 7 cycles, 6 = 6 cycles, 5 = 5 cycles, 4 = 4 cycles, Less than 3 = less than 3 cycles.

**Figure 4 cancers-14-02671-f004:**
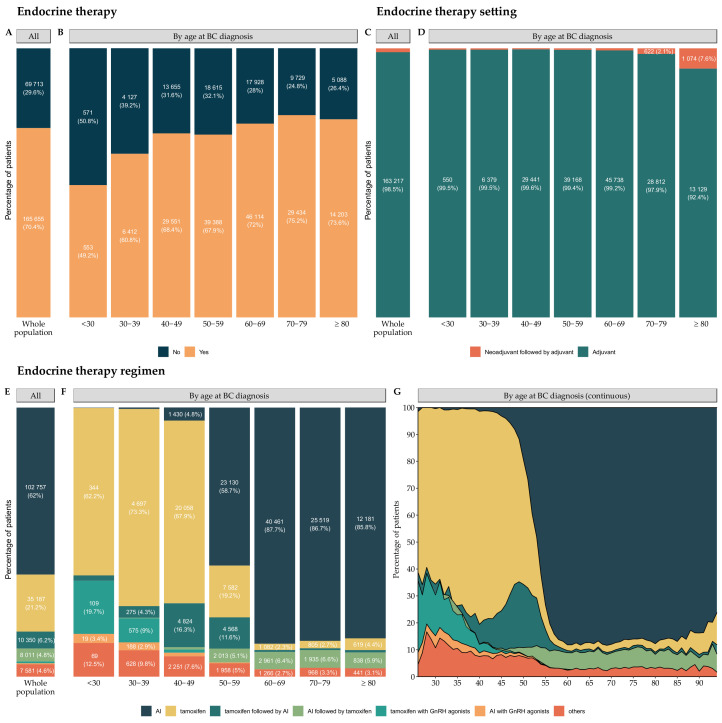
Endocrine therapy use, setting, and regimens, by age at BC diagnosis. (**A**) Endocrine therapy in the total population (*n* = 235,368); (**B**) Endocrine therapy by age class at BC diagnosis; (**C**) Endocrine therapy setting in total population (*n* = 165,655). Raw data for subgroups representing less than 2% of the total population are not displayed on the graph, to ensure readability: there were 2438 (1.5%) patients in the neoadjuvant followed by adjuvant group; (**D**) Endocrine therapy setting by age class at BC diagnosis. Raw data for subgroups representing less than 2% of the corresponding age class are not displayed on the graph, to ensure readability. The values not displayed are: for the <30 years old class: *n* < 10, 30–39 years old: *n* = 33 (0.5%), 40–49 years old: *n* = 110 (0.4%), 50–59 years old: *n* = 220 (0.6%), and 60–69 years old: *n* = 376 (0.8%) for the neoadjuvant followed by adjuvant group; (**E**) Endocrine therapy regimen for the total population (*n* = 165,655). Raw data for subgroups representing less than 2% of the total population are not displayed on the graph, to ensure readability: there were 1064 (0.6%) patients in the tamoxifen with GnRH agonists group, and 705 (0.4%) patients in the AI = (aromatase inhibitor) in combination with agonist group; (**F**) Endocrine therapy regimen by age class at BC diagnosis. Raw data for subgroups representing less than 2% of the corresponding age class are not displayed on the graph, to ensure readability. The values not displayed are: for the <30 years old group: n < 10 and 30–39 years old: *n* = 35 (0.5%) for the AI group. The values by age class for the tamoxifen followed by AI group are: for the <30 years old group: *n* = 11 (2%), 60–69 years old: *n* = 341 (0.7%), 70–79 years old: *n* = 207 (0.7%), and ≥80 years old: *n* = 124 (0.9%). The values by age class for the AI followed by tamoxifen group are: for the 30–39 years old group: *n* = 14 (0.2%) and 40–49 years old: *n* = 250 (0.8%). The values by age class for the tamoxifen with GnRH agonists group are: for the 40–49 years old group: *n* = 362 (1.2%) and 50–59 years old: *n* = 18 (0%). The values by age class for the AI with GnRH agonists group are: 40–49 years old: *n* = 376 (1.3%), 50–59 years old: *n* = 119 (0.3%), and 60–69 years old: *n* < 10. All other missing labels are 0; and (**G**) Endocrine therapy regimen by age at BC diagnosis. The cohort is restricted to patients aged from 26 to 94 years (*n* = 165,489); Abbreviations: BC = breast cancer and AI = aromatase inhibitor.

**Figure 5 cancers-14-02671-f005:**
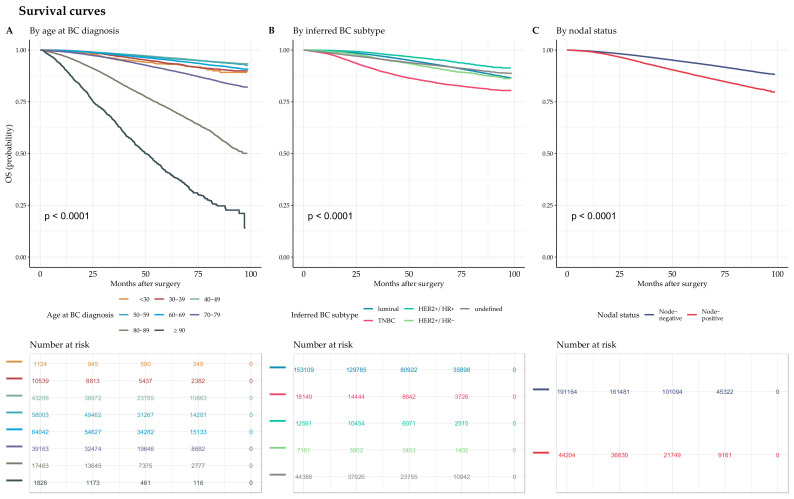
Unadjusted Kaplan-Meier survival curves. (**A**) Unadjusted Kaplan-Meier survival curves for the association between overall survival (OS) and age class at BC diagnosis. (**B**) Unadjusted Kaplan-Meier survival curves for the association between OS and inferred BC subtype. (**C**) Unadjusted Kaplan-Meier survival curves for the association between OS and nodal status.

**Table 1 cancers-14-02671-t001:** Baseline characteristics of the French Early Breast Cancer Cohort.

Variable	Class	All
*n* =		235,368
Age at diagnosis (years)		60.0 (19.0)
Age at diagnosis (years, classes)	(0–30)	1124 (0.5%)
	(30–40)	10,539 (4.5%)
	(40–50)	43,206 (18.4%)
	(50–60)	58,003 (24.6%)
	(60–70)	64,042 (27.2%)
	(70–80)	39,163 (16.6%)
	80+	19,291 (8.2%)
Inferred BC subtype	Luminal	153,109 (65.1%)
	TNBC	18,149 (7.7%)
	*HER2*^+^/HR^+^	12,561 (5.3%)
	*HER2*^+^/HR^−^	7161 (3.0%)
	Undefined	44,388 (18.9%)
Nodal status	Node negative	191,164 (81.2%)
	Node positive	44,204 (18.8%)
Surgery	No	0 (0%)
	Yes	235,368 (100%)
	Surgery type	
	*Partial Mastectomy*	*173,173 (73.6%)*
	*Mastectomy*	*62,195 (26.4%)*
	Axillar surgery	
	*No*	*38,231 (16.2%)*
	*Yes*	*197,137 (83.8%)*
Chemotherapy	No	145,116 (61.7%)
	Yes	90,252 (38.3%)
	Setting	
	*Neoadjuvant*	*15,627 (17.3%)*
	*Adjuvant*	*72,939 (80.8%)*
	*Both*	*1686 (1.9%)*
	Regimen *	
	*Anthracyclines*	*773 (0.8%)*
	*Anthracyclines/Docetaxel*	*13,897 (15.1%)*
	*Anthracyclines/Paclitaxel*	*21,605 (23.5%)*
	*Docetaxel*	*4974 (5.4%)*
	*Paclitaxel*	*2869 (3.1%)*
	*Other*	*6441 (7.0%)*
	*Unknown (Anthracyclines or Docetaxel)*	*41,379 (45.0%)*
Targeted therapy	No	215,646 (91.6%)
	Yes	19,722 (8.4%)
	Setting	
	*Neoadjuvant followed by adjuvant*	*4424 (22.4%)*
	*Adjuvant*	*15,298 (77.6%)*
	Regimen	
	*Trastuzumab only*	*19,289 (97.8%)*
	*Pertuzumab +/− trastuzumab ***	*433 (2.2%)*
Radiotherapy	No	34,683 (14.7%)
	Yes	200,685 (85.3%)
	Setting	
	*Neoadjuvant*	*323 (0.2%)*
	Adjuvant	*200,180 (99.7%)*
	*Both*	*182 (0.1%)*
Endocrine therapy	No	69,713 (29.6%)
	Yes	165,655 (70.4%)
	**Setting**	
	*Neoadjuvant followed by adjuvant*	*2438 (1.5%)*
	*Adjuvant*	*163,217 (98.6%)*
	**Regimen**	
	*AI*	*102,757 (62.0%)*
	*Tamoxifen*	*35,187 (21.2%)*
	*Tamoxifen followed by AI*	*10,350 (6.2%)*
	*AI followed by tamoxifen*	*8011 (4.8%)*
	*Tamoxifen in combination with agonist*	*1064 (0.6%)*
	*AI in combination with agonist*	*705 (0.4%)*
	*Others*	*7581 (4.6%)*

Abbreviations: HR^+^ = hormone receptor-positive; HR^−^ = hormone receptor-negative; *HER2*^+^ = human epidermal growth factor receptor 2-positive; TNBC = triple-negative breast cancer subtype; AI = aromatase inhibitor; *: chemotherapy regimens are displayed for the 91,938 (72,939 + 15,627 + 2 × 1686) chemotherapy settings identified. **: 3 of 19,722 with targeted therapy received pertuzumab only.

## Data Availability

The data presented in this study are available on request from the corresponding author.

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
