# Peer review of "The French Early Breast Cancer Cohort (FRESH): A Resource for Breast Cancer Research and Evaluations of Oncology Practices Based on the French National Healthcare System Database (SNDS)"

_cancers, 2022, doi:10.3390/cancers14112671_

Round 1

Reviewer 1 Report

I have following questions/remarks:

  1. unfortunately, no information about the histological characteristics of the tumor was available from the database. These histological characteristics are of greatest importance for therapy strategy in early breast cancer and concluding the histological subtype by treatment strategy doesn´t result in valid data, as demonstrated by 18.9% unidentifiable patients. Important questions, i.e. how many HR+ BC patients receive endocrine therapy, can not be answered. For future projects collection of histological results should be considered.
  2. In paragraph 2.2, ethics and data protection were presented. You mentioned that personal data were kept confidential by pseudoanonymizing the data. Can you explain how written consent was not applicable and required- is that per national cancer database, all patient data an be publicly analyzed by law?
  3. with regards to chemotherapy regimens nearly half of the patients were docomented as unknown regimen. Could you explain, why you assume that these unknown regimens mainly contain anthracyklines and or docetaxel, as you stated in paragraph 2.4.3?
  4. you stated that your data are consistent to the data from other sources (i.e. NCCN, SEER) in many aspects, did you find any major differences?

Author Response

Dear Professor Dr. Samuel C. Mok,

Please find enclosed our revised manuscript entitled “The French Early Breast Cancer Cohort (FRESH): a resource for breast cancer research and evaluations of oncology practices based on the French National Healthcare System Database (SNDS)”.

We would like to thank the Editor for this conditional acceptance and for kindly accepting to review our revised manuscript. We also thank the Editor and the reviewers for the important comments and the essential revisions they suggested. We feel that revisiting and reshaping extensively the manuscript according to the reviewer’s recommendations greatly improved its readability.

All changes required by reviewers #1 and #2 were made as suggested.

Major modifications are as follows:

  • We incorporated mental health status diagnosis as a perspective of the FRESH cohort use and provided some preliminary analyses on the evolution of psychotrop use (anxiolytics, hypnotic, antidepressants, antipsychotics) (#2); further studies are suggested.
  • We commented our results on the 18.9% percentage of undefined tumors found, which is consistent with the expected rates including DCIS, invasive tumors of small size without any additional systemic treatment, patients with hormone receptor positive tumors and/or HER2-positive tumors who refused or who were not offered the corresponding treatment (#1).
  • We analyzed major differences with SEER data and made hypotheses (#1):
    • For the age at BC diagnosis, SEER patients were older than in FRESH cohort, probably due to differences in tumor biology, notably BC subtype.
    • For BC biology, the difference for TNBC and HER2+, which were more frequent in the SEER cohort, might be explained by the different repartition of ethnicities in the USA; higher lymph node involvement at BC diagnosis in SEER cohort could be explained by delayed BC diagnostics as a proportion of patients do not have health insurance and may be diagnosed later.
    • For BC treatments, SEER patients more often experienced radical surgical and axillar surgery probably because of the lower proportion of radiotherapy in US, the cost of radiation treatments and/or the low density of radiation facilities across the country. Chemotherapy use was similar in the two cohorts.

In addition, we answered all the questions asked by reviewer #1, #2:

  • We explained that we inferred chemotherapy regimens from the intervals between chemotherapy sessions when the molecule used was not recorded (#1).
  • The proportion of BC subtype is in accordance with existing evidence in large scale studies (SEER, Swedish Cancer Register, Norway Cancer Register, Netherlands Cancer Registery) (#1).
  • Ethics and data protection were clarified in accordance with French regulations applicable to the SNDS data (#1).

We thank the reviewers again for their significant contribution that substantially improved the quality of our study and the clarity of our manuscript. We are looking forward to hearing from you.

Sincerely,

Dr Anne-Sophie Hamy MD, PhD

Reviewer 2 Report

First of all, I would like to note that we are dealing with an original study, the logic of which and the presentation of the main results is reliable and reasonable.
The advantage is the sample of respondents - case studies of 235,368 patients. The fact that the authors applied 10 filters for patient inclusion in the cohort deserves a positive assessment. Thus, there are clear criteria for differentiation.
The only thing missing in the article is the analysis of the mental status of patients and information on the assessment of their psychosocial rehabilitation. However, this may be a prospect for future authors' research.

Author Response

(The authors gave the same response as above.)
